# A browser-based tool for visualization and analysis of diffusion MRI data

Jason D. Yeatman [1,2], Adam Richie-Halford[3], Josh K. Smith[4], Anisha Keshavan[1,2,5] & Ariel Rokem [5]

Human neuroscience research faces several challenges with regards to reproducibility. While scientists are generally aware that data sharing is important, it is not always clear how to share data in a manner that allows other labs to understand and reproduce published findings. Here we report a new open source tool, AFQ-Browser, that builds an interactive website as a companion to a diffusion MRI study. Because AFQ-Browser is portable—it runs in any web-browser—it can facilitate transparency and data sharing. Moreover, by leveraging new web-visualization technologies to create linked views between different dimensions of the dataset (anatomy, diffusion metrics, subject metadata), AFQ-Browser facilitates exploratory data analysis, fueling new discoveries based on previously published datasets. In an era where Big Data is playing an increasingly prominent role in scientific discovery, so will browser-based tools for exploring high-dimensional datasets, communicating scientific discoveries, aggregating data across labs, and publishing data alongside manuscripts.

[1] Institute for Learning & Brain Sciences, University of Washington, Portage Bay Building, Box 357988, Seattle, WA 98195, USA. [2] Department of Speech and Hearing Sciences, University of Washington, Seattle, WA 98195, USA. [3] Department of Physics, University of Washington, Seattle, WA 98195, USA. [4] Department of Chemical Engineering, University of Washington, Seattle, WA 98195, USA. [5] eScience Institute, WRF Data Science Studio, University of Washington, Physics/Astronomy Tower (PAT), 6th Floor 3910 15th Ave NE, Seattle, WA 98195, USA. Correspondence and requests for materials should be addressed to J.D.Y. (email: jyeatman@uw.edu) or to A.R. (email: arokem@gmail.com)

Fueled by technical advances in modern web browsers and by the development of open source software libraries for interactive visualization, browser-based data visualizations have been playing an increasingly prominent role in communicating information on a wide range of topics. For example, news events are frequently accompanied by interactive maps, election results are reported in interactive plots, and research findings from a broad range of scientific research disciplines are displayed as interactive three-dimensional renderings[1]. JavaScript libraries for visualizing data in the web-browser[2,3] now rival most platform-specific packages in terms of plotting and rendering capabilities, and many scientific disciplines have further developed tools for the visualization of specific data types in the browser. In the field of neuroscience there are several different libraries devoted to visualization of brain imaging data. Examples include BrainBrowser[4], XTK[5], Mango[6] and Fiberweb[7], which provide application programming interfaces for programmers to create sophisticated applications that visualize three-dimensional brain structure with overlaid analysis results. These new tools are ushering in an era of Big Data in neuroscience and have laid the technical infrastructure for visualizing a breadth of commonly used medical imaging data types.

In the present work, we leverage these tools to develop the AFQ-Browser software that visualizes results from diffusion-weighted magnetic resonance imaging (dMRI) studies of human white matter. Even though many different methods for the analysis of dMRI data have been developed, there is broad agreement that tractometry[8–10], in which diffusion measurements are summarized along the length of fiber tracts, is a powerful analysis approach. There are currently two open source packages available to automate the process of identifying fiber tracts and quantifying tissue properties: Automated Fiber Quantification (AFQ[10]), which is implemented in MATLAB, and TRACULA[11], which combines FSL diffusion tools[12] and Freesurfer anatomy pipeline[13]. These software packages are widely used across clinical and basic science applications ranging from brain development and aging[14–19], autism spectrum disorders[20–22], major depressive disorder[23,24], head trauma[25–27], retinal disease[28], amyotrophic lateral sclerosis[29,30], surgical planning[31], and dyslexia[18,32,33]. Our present work focuses on designing a web-based graphical user interface (GUI) for tractometry. It confronts two major challenges in the study of human brain connectivity: (1) scientific reproducibility and (2) exploration of high-dimensional data. The narrow focus on tractometry allowed us to design a robust system that can be used by researchers without technical expertise in JavaScript and web visualization. Instead, we provide a command-line interface that allows researchers to visualize and explore data on their own computers and to publish results to the web.

Because AFQ-Browser is portable—it runs in any modern web-browser—it can be used to facilitate transparency and data sharing. The field of human neuroscience faces several specific challenges with regards to reproducibility[34,35]. Scientists are generally aware that data sharing is integral to reproducible research, but it is not always clear how to usefully share data. On one end of the spectrum, sharing raw data is often unwieldy[36], and reproducing the results from raw data requires access to the full series of computations that was used in the analysis. Computational complexity and data size can present a serious barrier that prevents scientists from attempting to reproduce published findings[37]. Moreover, the analysis of raw medical imaging data requires substantial domain expertise. This presents a barrier for researchers in computer science and statistics to apply innovations in their fields to the analysis of human brain data and to crosscheck the methodological assumptions of published work. On the other end of the spectrum, tables, graphs, and scatter plots that typically appear in journal articles reflect an author's interpretation of the data, but do not suffice for meaningful reproducibility of the results, or exploration of alternative theories.

Here, we propose that sharing dimensionally reduced portions of dMRI data, together with rich interactive data visualizations, lends itself not only to replication of original results, but to immediate and straight-forward extensions of these results, even in the hands of researchers in other disciplines. AFQ-Browser includes a function to publish the visualization and dimensionally reduced data to a publicly accessible website. Ideally, this intermediate form of data sharing would supplement the release of raw data, but it might also appeal to researchers who wish to communicate their findings more completely, but are not ready to release the full collection of raw data from an ongoing study, or worry about privacy concerns associated with raw data. AFQ-Browser automatically organizes dMRI data analyzed along tracts into tidy tables[38]. The software facilitates rapid publication of both the visualization and these data as an openly available website.

In designing a browser-based tool for sharing diffusion MRI data we further fill a growing void in the era of Big Data: the need for visualization tools to intuitively explore complex relationships in high-dimensional datasets. Data visualization and exploration plays an integral role in scientific inquiry, even beyond communicating results from statistical tests of an a priori hypothesis. High-dimensional datasets, such as Tract Profiles of white matter tissue properties measured with dMRI[10], in conjunction with behavioral and demographic measures in large samples of subjects, pose a fundamental challenge for data visualization. A solution pioneered by astronomy, genomics and other fields that were early to embrace Big Data was the development of tools implementing linked views of a data set, where interaction with a visualization of one dimension evokes a change in another visualization of the same data[39]. By interactively exploring the relationships among different dimensions of a dataset, a researcher can develop an understanding of the principles that characterize the system without specifying an a priori model of the complex relationships that are present in the high-dimensional data. Drawing inspiration from other disciplines that have already realized the power of linked view visualizations for exploring high-dimensional data, we present here a software tool that visualizes results from quantitative tractography analysis of dMRI data, and facilitates exploratory data analysis through the implementation of linked views of the data. By satisfying the need for both exploratory data analysis and data sharing, AFQ-Browser supports a virtuous cycle where public data are increasingly valuable and easy to share, and there are new opportunities to aggregate large datasets across laboratories.

## Results

**Generating new discoveries from old datasets**. Publishing data in a convenient format supports reproducibility and fuels new scientific discoveries. For example, examining the published data from Yeatman et al.[19] in a running instance of AFQ-Browser (http://YeatmanLab.github.io/AFQBrowser-demo), we can reproduce the previously reported finding that, in terms of mean diffusivity (MD), the arcuate fasciculus demonstrates more developmental change than the corticospinal tract (CST). When the sample is binned into three age groups, both the arcuate and CST show highly significant changes, but the magnitude of change between childhood and adulthood is larger for the arcuate than the CST (Fig. 1). By switching the plot to fractional anisotropy (FA) rather than MD, another effect, not reported in the original manuscript, can be observed. While the arcuate shows the expected pattern of results—FA values increase with

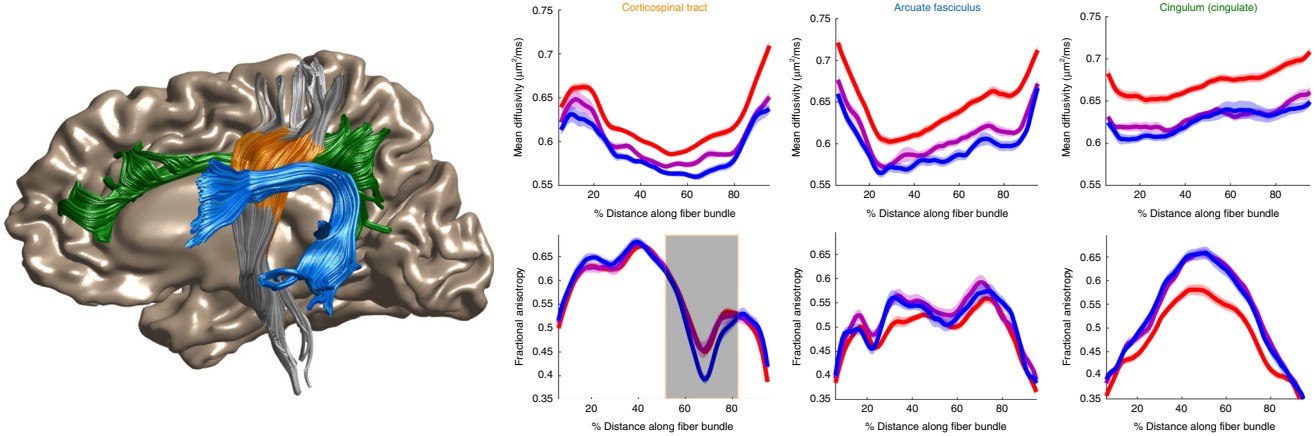

**Fig. 1** Development of the corticospinal tract, arcuate fasciculus, and cingulum. Tract profiles of mean diffusivity (top) and fractional anisotropy (bottom) are shown for the left hemisphere corticospinal tract (CST, orange), arcuate fasciculus (blue), and cingulum (green). Splitting the group by age, and selecting 3 bins, displays mean lines of three groups: 8–15 (red), 15–30 (purple), and 30–50 (blue). For the CST, there is a region that shows a decrease in FA with development, and this location of the tract is highlighted on the plot using the "brushable tracts" feature (shaded gray box). The linked view in the anatomy displays the portion of the CST that is brushed in the plot demonstrating that this effect occurs in the anatomical portion of the CST known as the centrum semiovale, adjacent to the arcuate fasciculus. This linked visualization provides a connection between the data plots and the 3D Anatomy. Data and MATLAB code are available at https://github.com/YeatmanLab/AFQ-Browser_data (see Figure2_Development.m) and running AFQ-Browser instance at: https://YeatmanLab.github.io/AFQBrowser-demo/)

development—the CST shows the opposite pattern of developmental change. For the CST, the three age groups have equivalent FA values for the first half of the tract, but adults have lower FA values than young adults or children between nodes 50 and 80 (Fig. 1). At first, this finding might seem counter-intuitive: FA typically increases with development as axons become more densely packed and myelinated. But in this case the developmental decline in FA occurs in the centrum semiovale, a portion of the Tract Profile where FA drops substantially due to crossing fibers. The developmental decline in FA is therefore likely to reflect development of the fiber tracts that cross through this portion of the CST, rather than changes in CST axons per se. This interpretation makes sense given that the superior longitudinal fasciculus, one of the tracts crossing through this region of the CST, is believed to continue developing into young adulthood. This interpretation of the developmental changes in FA in regions of crossing fibers offers some clarity to other reports of declining FA values in the young adult brain, but also requires a more thorough investigation in an independent dataset.

**Localizing white matter lesions in patients with multiple sclerosis.** Multiple sclerosis (MS) is a degenerative disease of the white matter characterized by progressive loss of myelin. Even though measures such as MD and FA are not specific to myelin, dMRI is still a promising technique for detecting and monitoring white matter lesions in MS and quantifying results from drug trials targeting remyelination[40]. DMRI is sensitive to aspects of the disease that are not detectable with conventional imaging methods (T1, T2, fluid-attenuated inversion recovery (FLAIR)). Quantitative comparisons between MS patients and healthy control subjects have demonstrated differences in diffusion properties within "normal appearing white matter", or regions that do not show obvious lesions on a conventional MRI image. In longitudinal studies, these regions with diffusion differences are likely to progress into lesions, indicating the sensitivity of dMRI for detecting early signs of the disease, and monitoring the benefit of drugs that aim to prevent the demyelination process[40–43].

One of the challenges for incorporating dMRI into clinical practice is the lack of user-friendly methods for visualizing results

in a quantitative manner. For clinical applications, group comparisons have limited utility, because ultimately the goal is to detect abnormalities and make diagnoses at the level of the individual. For example, in the data previously published by Yeatman et al.[19] and Mezer et al.[44], MD, radial diffusivity (RD), and FA values are significantly different in MS patients compared to controls for most tracts in the brain (https://YeatmanLab.github.io/AFQ-Browser-MSexample/). MD and RD show much greater sensitivity to group differences than FA: Fig. 2 shows group means and standard errors for MD, RD, and FA along the corticospinal tract, posterior callosum, inferior longitudinal fasciculus, and arcuate fasciculus.

Group comparisons demonstrate the sensitivity of the measure to the disease but do not provide diagnostic information about individual patients: each individual has tissue abnormalities in different parts of the brain, with some tracts showing diffusivity values in the normal range, others showing normal-appearing white matter on a T1, but abnormalities in terms of diffusion metrics, and other tracts displaying major lesions. AFQ-Browser provides a simple and intuitive method to quantitatively compare an individual's white matter tissue properties to normative data from healthy brains by plotting each individual's Tract Profile in comparison to the normative distribution of healthy brains (means and SDs, Fig. 3). When an individual is selected in the AFQ-Browser GUI, z-scores comparing that individual to the norms of each group are displayed above the individual's Tract Profile. Such a comparison can localize lesions to specific locations on a tract and quantify the extent of damage. Clinical data are a prime example of the utility of linked visualization: the links between quantitative plots of diffusion measures, tract anatomy, and subject metadata make it possible to quickly find a subject with a lesion, determine the location of the lesion and associate this information with clinical symptoms. While not as specific to myelin as other quantitative measurements such as R1[19,45–48], we find that MD and RD are highly sensitive to MS lesions. For example, the lesion shown in Fig. 8 of ref. [19] can be detected based on MD values that are 5.6 SD away from the norms, with a larger lesion in the left compared to the right occipital callosal connections (Fig. 3, subject_020). In this lesion, RD values are slightly more sensitive showing a z-score of 6.2 and

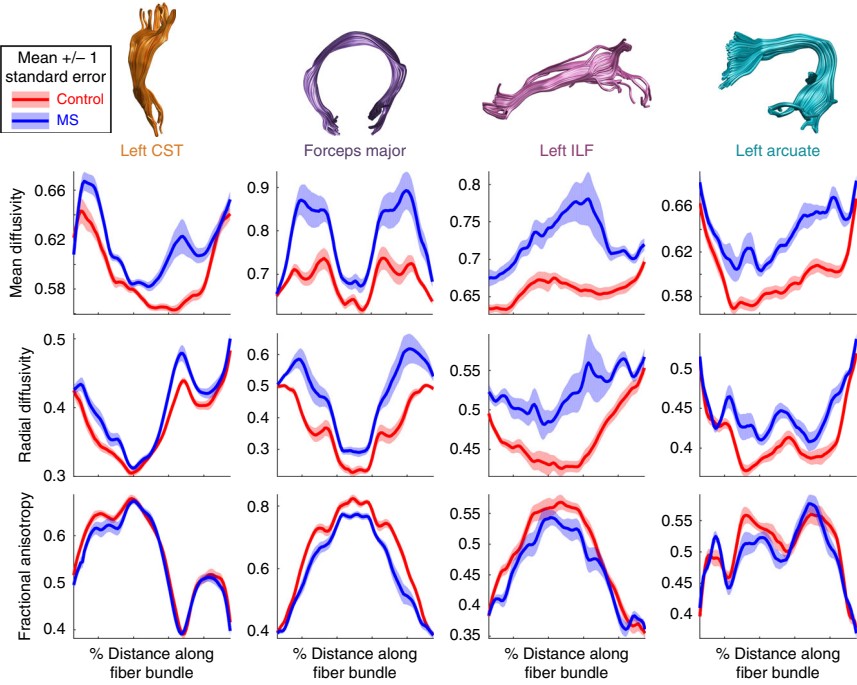

**Fig. 2** Group comparison between multiple sclerosis patients and healthy control subjects. We observe highly significant ($p < 0.001$) group differences in diffusion measures across many tracts. Mean diffusivity (top panel) and radial diffusivity (middle panel) show larger group differences than fractional anisotropy (bottom panel). Mean values +/- 1 standard error are shown for control subjects in red and multiple sclerosis (MS) patients in blue (see Figure3_4_MultipleSclerosis.m and https://YeatmanLab.github.io/AFQ-Browser-MSexample/)

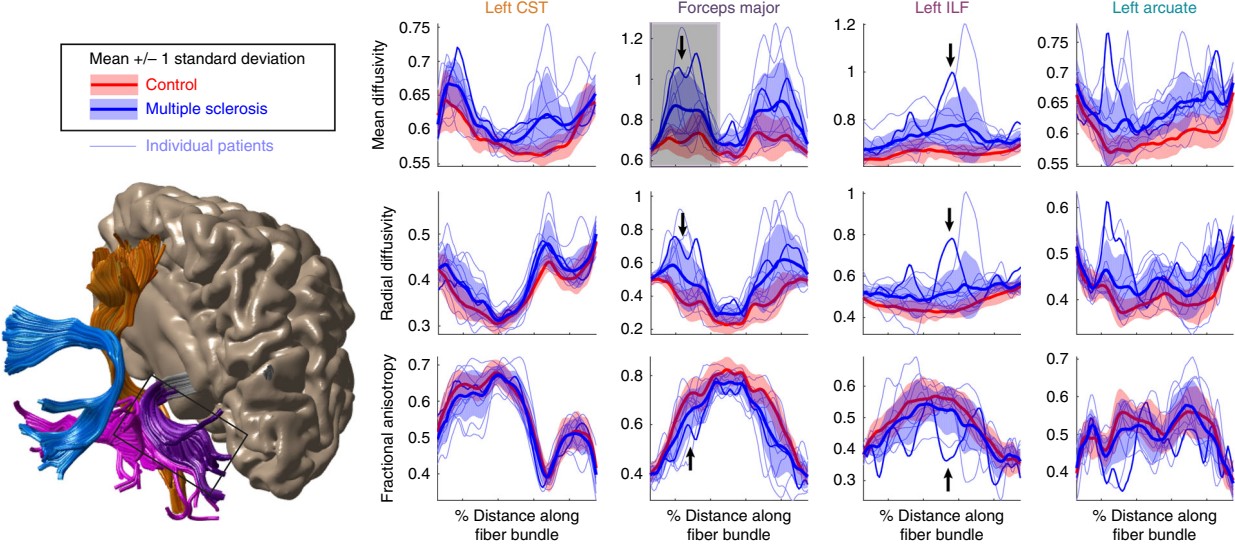

**Fig. 3** Localizing lesions in an individual's brain. Individual MS patients (light blue lines) are plotted against the normal distribution (mean +/- 1 standard deviation) of values in healthy control subjects. Lesions and diffuse abnormalities can be detected in individuals based on large deviations from the control subjects. The darker blue line is data from the patient shown in Figure 8 of Yeatman et al.[19]. By plotting standard deviations rather than standard errors, the large (>1 SD) difference between MS patients and control subjects is apparent, as are the large deviations of specific patients from the normal distribution (see Figure3_4_MultipleSclerosis.m and https://YeatmanLab.github.io/AFQ-Browser-MSexample/)

FA values are slightly less sensitive, with a $z$-score of $-3.3$ compared to healthy controls (Fig. 3). For this patient, the large lesion on the ILF was more than 10 SD greater than the controls in terms of MD and RD. As more clinical datasets are aggregated in public repositories there will be new opportunities to explore the sensitivity and specificity of this type of individual comparison.

**Detecting degeneration in amyotrophic lateral sclerosis.** Amyotrophic lateral sclerosis (ALS) is a neurodegenerative disease in which progressive degeneration of upper and lower motor neurons leads to atrophy, weakness, and loss of muscle control. The time-course of disease progression varies substantially across patients, with some showing rapid degeneration and others showing a sporadic or gradual decline. Due to the heterogeneous

presentation of clinical symptoms in ALS, early diagnosis can be challenging and the disease can go undetected in many patients until they present with severe symptoms. Hence, the development of quantitative and automated methods for diagnosis and disease monitoring has been a major focus within clinical neuroimaging research. Diffusion MRI holds promise as a tool to detect the early stages of neural degeneration and corroborate behavioral assessments. Group analyses have consistently demonstrated significant reductions in FA, increases in RD, and increases in MD in the corticospinal tract[29,49]. Group comparisons provide information about the average pattern of disease progression but ultimately the goal of clinical neuroimaging research is to develop techniques that have sufficient sensitivity and specificity to be applicable at the individual level. A recent study used AFQ and a random forest classifier to develop an automated diagnosis system to classify subjects as healthy or diseased based on dMRI measures[29]. They achieved 80% classification accuracy (cross-validated) based on Tract Profiles of the corticospinal tract and reported that FA and RD at the level of the cerebral peduncle and posterior limb of the internal capsule were the most informative diffusion properties. These effects can be visualized in AFQ browser by binning the subjects based on disease diagnosis (https://YeatmanLab.github.io/Sarica_2017/, Fig. 4). As reported by Sarica et al.,[29] the mean RD and FA values in this region of the CST are more than 1 SD different in ALS patients compared to controls (node 40, arrow, Fig. 4). Even though a multivariate classification strategy (random forests) is used to achieve good diagnostic accuracy, visualization of individual Tract Profiles demonstrates that a majority of patients (75%) deviated by more

than 1 SD from control values within the right CST at the level of the cerebral peduncle. Based on these data that were made publicly available through AFQ-Browser, we implement a series of computations in a Jupyter Notebook that reproduce the central findings, and a figure from the original work (https://github.com/YeatmanLab/AFQ-Browser_data/blob/master/AFQ-Browser_ALSexample/Reproducing-Sarica2017-Figure3.ipynb).

The goal of most clinical neuroimaging studies is to detect regions of the brain that are affected by the disease. While not a central focus of clinical research, there is also scientific importance to clearly establishing regions of the brain that are not affected by the disease. Based on the previously published data in Sarica et al.[29], we can investigate the specificity of the effects to the CST and determine whether there are any tracts that can be established as control regions not affected by the disease. We find that the CST is the only tract that shows large (>1 SD) differences between patients and controls in terms of RD and FA values. While there are a few regions that show small differences (depending on the statistical threshold), the specificity of the effects to the CST is striking. For example, many tracts including the forceps major and forceps minor of the corpus callosum and the left and right inferior fronto-occipital fasciculus show nearly identical distributions of values between patients and controls (Fig. 4).

**Removing barriers for interdisciplinary collaboration.** Statistics, machine learning, and data science are making impressive strides in the development of general-purpose methods for the interpretation of data across a variety of scientific fields[50]. One of

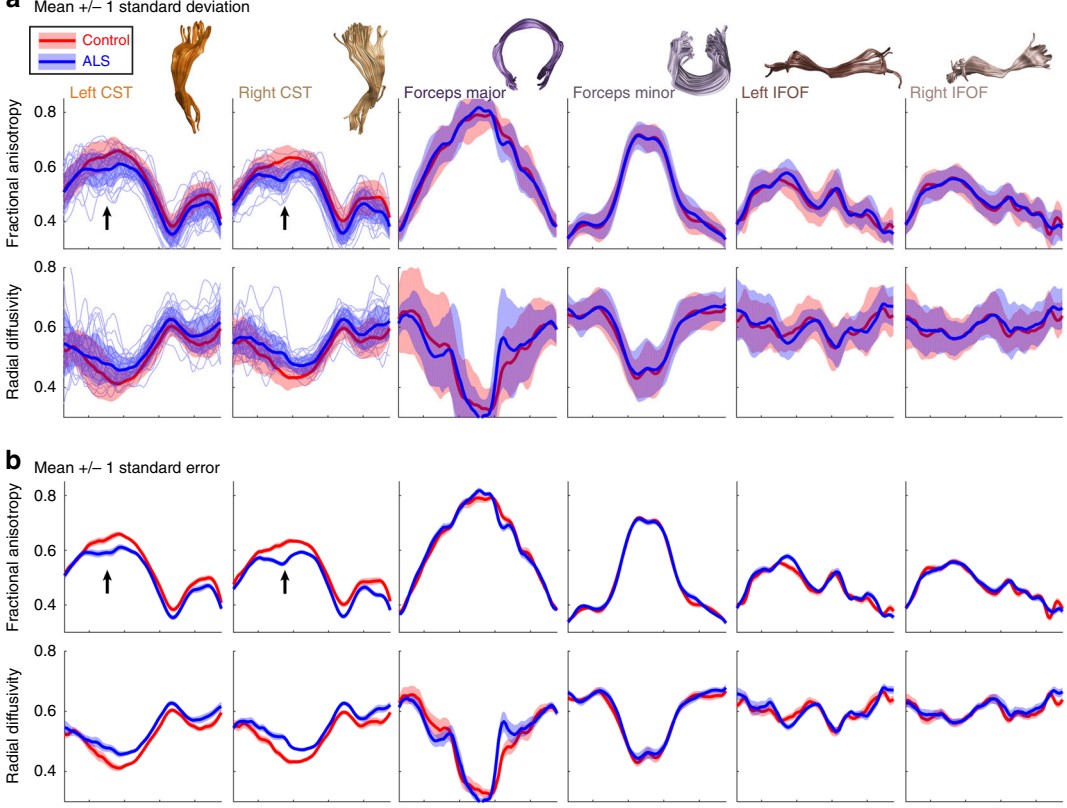

**Fig. 4** Amyotrophic lateral sclerosis patients show isolated degeneration of the corticospinal tract. **a** Means and standard deviations of FA and RD values are shown for ALS patients (blue) and control subjects (red). Individual patients are displayed as light blue lines for the CST. At the level of the cerebral peduncle, patients differ from controls by more than 1 standard deviation (black arrow). No other tracts show this large effect. **b** Means and standard errors are shown for ALS and control subjects to indicate regions of significant group differences. Group differences are relatively specific to the CST (see Figure5_ALS.m and https://YeatmanLab.github.io/Sarica_2017/)

the current barriers to a broader application of these methods is the extraction of useful analysis features from unstructured data sets that contain large, heterogeneous, noisy measurements, saved in obscure domain-specific or proprietary formats, that require special software, and arcane preprocessing steps. Brain imaging data are a paradigmatic case of this state of affairs: measurements are typically large, on the order of several gigabytes per individual, signal-to-noise ratio can be low, and differences in three-dimensional (3D) brain structure between individuals make naive image processing of the original measurement fraught. One of the major strengths of tractometry is that it extracts features from brain imaging data based on domain-specific knowledge: quantitative measurements of tissue properties for well-defined anatomical segments of the major white matter connections in an individual's brain[51]. This reduces the dimensionality of the data substantially, while still retaining rich, complex information about an individual's neuroanatomy.

AFQ-Browser provides these domain-relevant features in a format that will be familiar to many machine learning and statistics practitioners: tables with observations as rows, and variables as columns. This format, known as tidy data[38], is the universal exchange format of data science. The data are converted by the AFQ-Browser software and stored in ubiquitous text-based formats: CSV and JSON files. Separate tables are available for node-by-node estimates of the diffusion properties along the length of the fiber groups, and for the subject metadata, and these tables can be merged in an unambiguous manner through a shared subject ID variable. These files can be read using the standard data science tool-box: Software libraries such as the Python pandas library[52], or using the R statistical language[53]. Once data are read into tables, data processing and visualization with tools such as Seaborn (https://seaborn.pydata.org/) or ggplot (http://ggplot2.org/) are also straightforward. Furthermore, very few steps are required to apply machine learning techniques to the data, using tools such as the scikit-learn library[54], and results such as classifier weights can be easily interpreted with respect to known brain anatomy. An example of such an analysis is presented in Fig. 5, using the same data as in Fig. 4.

To lower the barrier to data use even further, we integrate AFQ-Browser with the Binder service (https://elifesciences.org/labs/8653a61d/introducing-binder-2-0-share-your-interactive-research-environment): this web-service allows visitors to an AFQ-Browser instance to click a button that takes them to a cloud-based interactive Jupyter notebook, with the data from this instance accessible, and Seaborn, Pandas, and scikit-learn already installed. Thus, visitors to an AFQ-Browser site can immediately start analyzing the data, even without downloading anything to their personal computer, or installing any additional software.

## Discussion

We have developed a new visualization tool for the quantitative analysis of diffusion MRI data in the web browser. The goals of this work were twofold: first, to support scientific reproducibility by removing barriers to public data release and, second, to capitalize on new technologies for linked visualization that facilitate exploratory data analysis. AFQ-Browser makes it possible to create an interactive website of companion data for a manuscript with a single command (afqbrowser-publish). While, ultimately, we advocate for releasing all the raw data and analysis code associated with published work[36], we also maintain that releasing derived measures (Tract Profiles) is a major step in the right direction and will allay the concerns that many scientists feel about giving up control of difficult to collect data sets. Ideally, this practice will serve as a stepping-stone to further data and code sharing.

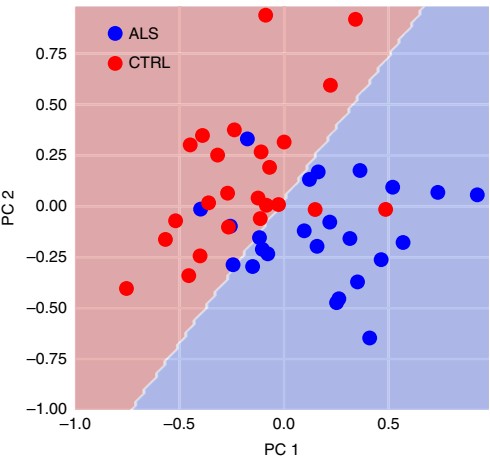

**Fig. 5** Classification of ALS patients based on FA in the corticospinal tracts. The Tract Profiles in the two CSTs are submitted to a Principal Components Analysis—the first two PCs form the dimensions of this plot (accounting for about 50% of the variance in the data). The data are separately used to train a support vector machine classifier, with a polynomial kernel. The classification boundary is shown here in the space of first two PCs. This classifier performs at 88% accuracy (cross-validated) in discriminating patients from controls. The Jupyter Notebook containing all steps of the analysis is shared here: https://github.com/YeatmanLab/AFQ-Browser_data/blob/master/AFQ-Browser_ALSexample/Figure6.ipynb

An additional benefit of releasing derived measures is that readers of a manuscript can easily explore dimensions of the data that were not reported in the publication. For example, it is not feasible to report results for every possible diffusion metric, and it is common for a manuscript to focus on a single metric. In our previous work[19] we only reported modeling results for MD, R1, and MTV. A reader that is left wondering whether other metrics (e.g., RD and AD) would show the same pattern of results can now quickly answer this question through the companion website (http://YeatmanLab.github.io/AFQBrowser-demo). Not only is a companion website more feasible than a supplement that includes every potential analysis, but also through AFQ-Browser researchers can extend published work and make new discoveries. For example, we have made three observations that extend the findings reported in published datasets: (1) in regions of crossing fibers there are developmental declines in FA (Fig. 1); (2) MS lesions can be detected in an individual, and localized on a tract, based on RD or MD but not FA (Fig. 3); (3) white matter degeneration in ALS is highly specific to the corticospinal tract and many cortical association tracts are largely unaffected by the disease (Fig. 4). While each of these discoveries is only an incremental contribution to what was reported in the original work, we contend that having datasets openly available online, with tools that facilitate data exploration, will fuel important new discoveries in human neuroscience.

We are not the first to create interactive web-based visualizations to accompany a manuscript. For example, the Allen Brain Institute has built a powerful GUI to explore large, multimodal genomics and physiology datasets (http://casestudies.brain-map.org/celltax). Friederici et al.[55] built an interactive brain viewer to accompany a review paper on the neuroanatomy of language so that readers could explore anatomy in a more detailed manner than is possible in a static figure (http://onpub.cbs.mpg.de/index.html). The BigBrain project[56] has released a high-resolution atlas of the human brain histology that can be navigated based on custom WebGL code (https://bigbrain.loris.ca). Huth and colleagues used pyCortex[57] to build an interactive website to

accompany recent work[58] on the structure of semantic maps in the human brain (http://gallantlab.org/huth2016/). There are numerous examples of beautiful interactive websites that labs have designed to accompany key studies, and interact with landmark datasets. However, these major achievements in browser visualization are isolated to a few labs with high technical capabilities and the willingness to invest the time and resources required to design a custom website for a publication. AFQ-Browser fills an important gap by removing these constraints: a website can be published by running a single command (afq-browser-publish) in a software package that can be installed automatically on any machine with Python (pip install AFQ-Browser), the website is hosted for free through GitHub Pages, and the underlying data are permanently stored in afqvault (http://afqvault.org). Thus, even labs with minimal resources and technical capabilities can communicate important scientific findings in an interactive format.

In building a tool like AFQ-Browser, should we worry that we are supporting scientific transparency at the expense of artificially diminishing $p$-values? In other words, is exploratory analysis of public datasets at odds with hypothesis testing? The statistician John Tukey[59] coined the term "exploratory data analysis" to describe the process of data analysis through iterative processing, probing, and visualization of datasets. Tukey argued for a sharp distinction between exploratory and confirmatory data analysis (or hypothesis testing), and posited that scientists should strive to obtain multiple datasets allowing them to explore a high-dimensional system, and develop hypotheses through exploratory data analysis, before performing the formal statistical tests to confirm or reject their hypothesis based on an independent dataset. Traditionally, the field of cognitive neuroscience has approached data analysis with the goal of testing specific hypotheses. Thus, experiments and data collection are designed with a hypothesis in mind, and data analysis involves computing statistics to formally test this hypothesis. In hypothesis-driven science, data visualization is often viewed as separate from the scientific investigation, but in complex systems, with non-linear relationships, exploratory data analysis and visualization can be essential for clarifying patterns that might be obscured in a conventional statistical analysis[60]. With new imaging techniques, and large-scale data collection efforts, the field of human neuroscience sits at a transition point, where exploratory data analysis and data-driven discovery is becoming appreciated as increasingly important. Other scientific fields such as astronomy and genomics that have embraced Big Data have discovered the critical role that data visualization can play in developing new theories[39]. As the field of human neuroscience transitions to an era of Big Data, tools like AFQ-Browser will become increasingly important as a way for scientists to interact with large datasets. As datasets grow, so will the importance of tools that can operate in the same manner on data stored on a personal computer in a laboratory, or on remote datasets stored in the cloud. Browser-based GUIs can fill this growing need.

However, we might also worry that in developing tools like AFQ-Browser, we are supporting reproducibility and data mining at the expense of "p-hacking"[61–63]. This is a valid concern and highlights the need for our standards on scientific rigor to evolve with the changing landscape of Big Data. For example, a lab might typically only conduct a limited number of statistical tests and, ideally, would correct $p$-values for each statistical test that was performed (not just the tests that were reported in the manuscript). But exploratory data analysis involves examining many possible processing pipelines and relationships between variables in a system[59]. The strength of tools like AFQ-Browser is the ease of exploring large datasets to identify relevant dimensions, and make data-driven discoveries that suggest a new

hypothesis to test in future work. Data exploration is a critical component of hypothesis generation, and data mining tools should not be discarded over worries of p-hacking. But thoughtful consideration of statistical concerns is also paramount. Drawing a distinction between exploratory and confirmatory data analysis allays concerns over biased $p$-values by defining the central role of replication in scientific discovery. An observation that emerges from exploratory data analysis should be confirmed in an independent dataset. As more datasets become publicly available, confirmatory data analysis and independent replication will become standard practice. Tools like AFQ-Browser facilitate this goal of aggregating many independent datasets. Finally, Big Data should not be viewed as a replacement for small and careful, hypothesis-driven investigations within a single laboratory. The field should strive for a balance between the innovative data-driven discoveries that can emerge from large public datasets, and the careful, targeted scientific investigations that a lab can undertake to definitively test a specific hypothesis.

## Methods

**The AFQ-Browser software**. AFQ is a software package for quantitative analysis of white matter fiber tracts[10]. The AFQ software is a fully automated pipeline that takes in diffusion MRI data and returns Tract Profiles of diffusion properties (or other quantitative MRI parameters) sampled along the trajectory of 24 major white matter fiber tracts (i.e., tractometry[8,9]). Fiber tracts are identified in an individual's native space, and the diffusion properties are sampled at points along the trajectory of each tract, thereby representing the data for each tract as a vector of measurements. For groups of subjects, data for a tract are represented by a matrix of values where each row corresponds to a subject and each column corresponds to a node along the tract. This pipeline can be thought of as a dimensionality reduction technique, whereby the data from hundreds of thousands of voxels get summarized in terms of features (fiber tracts) that have a known anatomy, and are important for specific aspects of cognitive function. Based on this dimensionality reduction and alignment into the individual participant's anatomy, groups of subjects can be compared in terms of these features, individuals can be compared to groups, and supervised and unsupervised learning techniques can be applied to link white matter biology to cognition in health and disease. But even this lower-dimensional view of the diffusion data can become unwieldy as datasets grow larger, and as there is an increasingly complex collection of subject metadata characteristics (e.g., behavioral measures, demographics, disease state, etc.) that might be linked to the underlying biological measurements. Hence, data visualization that allows for linked views across different dimensions of the dataset is essential.

AFQ-Browser takes the output of the AFQ MATLAB tractometry pipeline, and generates a browser-based visualization of the results. The AFQ MATLAB analysis pipeline produces a standard AFQ object, stored as a MATLAB .mat/hdf5 file. This file contains a structure array data-structure, with Tract Profiles for all the diffusion properties that were calculated from the dMRI data, for all tracts and subjects. The AFQ file also contains a field for metadata: subject-level characteristics, such as age, clinical diagnosis, or scores on psychometric tests are saved in this field. A command line function, afqbrowser-assemble, extracts all this information from the AFQ .mat file and writes out the hierarchically nested structured array as a series of .csv and .json files, stored in tidy formats[38]. This command line application then organizes the various AFQ-Browser files into a fully functioning AFQ-Browser website: a template of HTML and JavaScript scripts, and CSS styling are arranged into the appropriate folder structure, and the data are placed in a data folder from which the application files read it into the browser. A second command line function, afqbrowser-run, launches a static web-server on the user's computer with AFQ-Browser running for this dataset. Running a static web-server is required to access locally stored data files. Navigating a web browser to the returned URL (defaulting to https://localhost:8080) will open the visualization.

Even though AFQ-Browser was designed specifically to interact with the AFQ software, there are many other approaches to deriving Tract Profiles of diffusion properties, and we have designed AFQ-Browser to be broadly compatible with other software packages. The afqbrowser-assemble function also accepts the output of a group analysis in TRACULA (stats folder, see https://surfer.nmr.mgh.harvard.edu/fswiki/FsTutorial/Tracula for TRACULA documentation). Moreover, the data format used by AFQ-Browser is not specific to either of these software packages, and it is extensively documented (https://YeatmanLab.github.io/AFQ-Browser/dataformat.html) such that any other software pipeline can easily leverage the afqbrowser-run and afqbrowser-publish function to analyze and publish data, if it is formatted according to these specifications.

**Linked visualization**. The browser-based GUI has four panels (Fig. 6: (a) BUNDLES ; (b) ANATOMY; (c) BUNDLE DETAILS; and (d) SUBJECT METADATA). The visualization is linked across the panels in four ways.

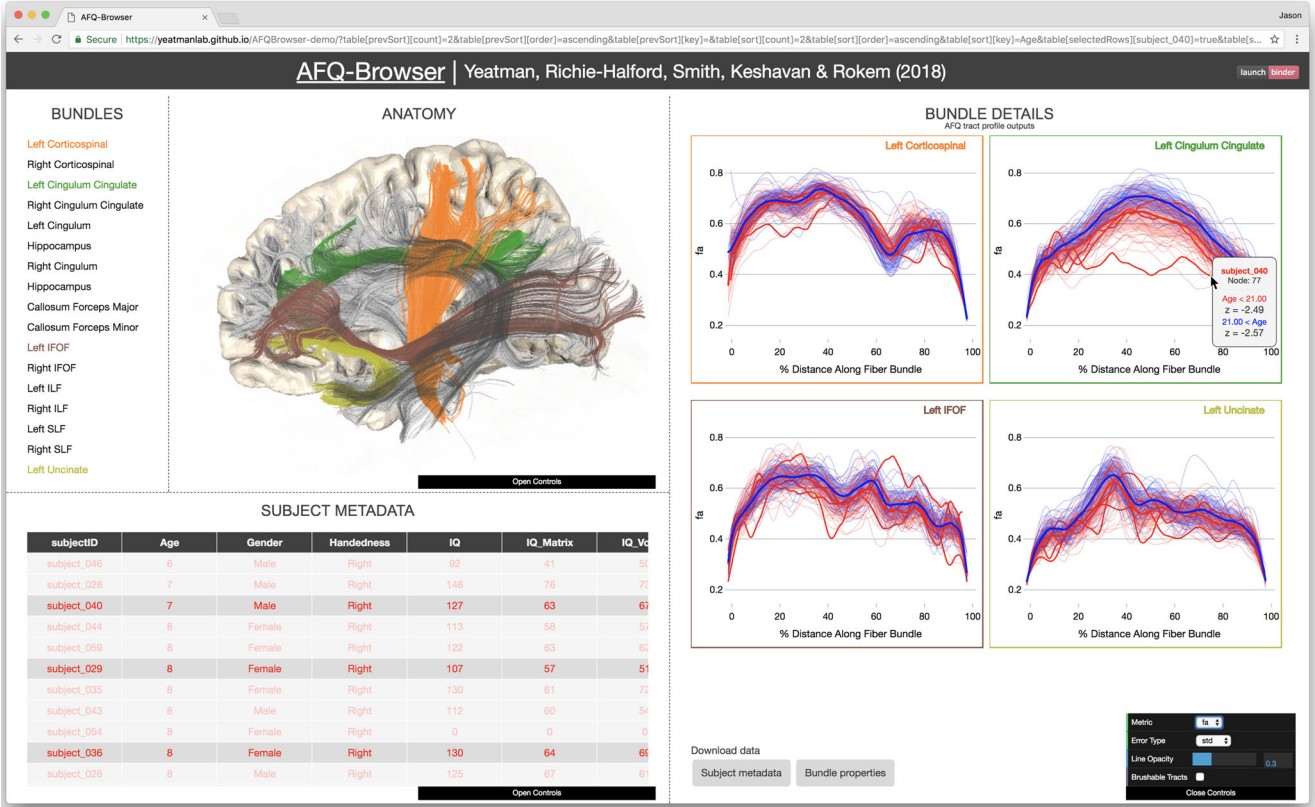

**Fig. 6** AFQ-Browser. The BUNDLES panel displays the names of the tracts and the colors are linked to the ANATOMY and BUNDLE DETAILS panels. Selecting a tract in the BUNDLES or ANATOMY panel will display the Tract Profile in the BUNDLE DETAILS panel. Selecting an individual subject's Tract Profile will highlight that subject in the SUBJECT METADATA panel. Selecting a column of SUBJECT METADATA groups subjects based on this measure. In the example, subjects are grouped based on age and means and standard deviations are shown in the BUNDLE DETAILS panel. When the mouse hovers over a selected subject's Tract Profile, z-scores are displayed for that subject relative to the group

First, color is used to identify each fiber tract (here referred to as "Bundle") across the Bundle List, 3D Brain visualization, and Bundle Details plots. We use the categorical Tableau-20 color scheme (https://www.tableau.com/about/blog/2016/7/colors-upgrade-tableau-10-56782). Clicking on a tract in the Bundle List, or 3D Brain, will highlight that tract in both panels and open up the corresponding line plot showing diffusion properties of that tract for each subject.

Second, the Tract Profiles from each individual subject in the Bundle Details panels are linked to their metadata. Selecting a Tract Profile (line) in the Bundle Details plot will highlight that subject's row of the metadata table, and selecting a row of the metadata table will highlight that subject's Tract Profile in the plot. A subject of interest can be selected based on their metadata to visualize their Tract Profiles relative to the group of other subjects, or a Tract Profile of interest can be selected to compare their metadata against the group of other subjects. Z-scores are displayed on mouse hover over for selected Tract Profiles, providing a statistical summary of an individual's data relative to the group.

Third, columns in the Metadata table are linked to mean lines in the Bundle Details plots. Clicking a column will sort the metadata table based on the data in that field, and subjects will be divided into $N$ groups by binning the data (the number of groups can be defined in a control bar). Each bin will be assigned a color and this color will be used for the rows of the metadata table, the mean lines in the Bundle Details plot, and the individual subject lines in the plot. Each time a new column in the metadata is selected, the mean lines are updated in the plot, and the rows are sorted and colored appropriately in the metadata table. This feature provides an efficient tool to slice a large data set across different dimensions, examine how different subject characteristics relate to diffusion measures, determine subjects that are outliers within a group, and determine how different manners of grouping produce changes across different white matter fiber tracts. Subject z-scores are updated based on the grouping.

Fourth, the spatial dimension (x-axis) of the Bundle Details plots is linked to the fiber tracts in the 3D brain visualization. Manual selection (brushing[64]) of a range of nodes in the Bundle Details plot, enabled by toggling on the brushable tracts feature in a control bar, highlights the corresponding region of the fiber tract in the 3D brain. This feature allows a user to link statistics, group differences, or quantitative comparisons of an individual subject back to their brain anatomy.

**Publishing data for reproducible science.** A single command, afqbrowser-publish, packages the entire website, including both data and visualization into a git

repository, and uploads this repository to GitHub (https://github.com). This script automatically creates a website with these data, hosted on the repository's "GitHub Pages" website, so that it can be viewed by anyone through a web-browser. The published website also includes a link that allows users to download the .csv files that contain the information that is displayed, for additional computational exploration through other tools (e.g., by reading the data into scripts that implement machine learning algorithms). The only requirement is that the user has a GitHub account and afqbrowser-publish will create the public repository, build the webpage, and launch the web server through GitHub.

To create a centralized index of public AFQ-Browser instances, and to aggregate data across studies, we have implemented a centralized database at http://afqvault.org, akin to the NeuroVault (https://neurovault.org) database for functional MRI derivatives[65] (and capitalizing on the infrastructure from other open source database projects in the field[65,66]). In addition to launching an AFQ-Browser instance on GitHub, the afqbrowser-publish command also commits the data to the afqvault database. This database stores all the data and the parameters from the AFQ object (including scan parameters if these were entered).

Long-term preservation of the data is important and, because GitHub does not guarantee long-term storage, we suggest using another service to ensure that the data are accessible in perpetuity. A one-click solution is provided through Zenodo (http://zenodo.org/)[34], a website developed by CERN specifically to support long-term preservation of data and other research products. Zenodo can be used to mint a persistent digital object identifier (DOI) for GitHub repositories. Other solutions include institutional repositories, to which users of AFQ-Browser can upload their data. We do not intend to enforce one solution or another, and we provide users with maximal control over this process. However, to provide users with information about the reasons to pursue long-term preservation, and about one way they could approach this, we provide documentation for issuing a DOI through Zenodo (https://YeatmanLab.github.io/AFQ-Browser/long_term_preservation.html).

**Saving the browser state.** Reproducing results that are generated by a GUI can be problematic since figures are generated based on a series of user inputs (i.e., mouse clicks and key presses). To solve this problem, we record the series of interactions as a query string, and append the URL with each user input to AFQ-Browser. By copying or bookmarking the URL, a user can save and re-open a specific state of

AFQ-Browser. Hence, a discovery made through a series of operations in the GUI is recorded in the URL and can be communicated (and reproduced) without a lengthy description of the series of user inputs.

**Integration with Binder to support extensible computations**. While AFQ-Browser supports flexible exploration of the data over many dimensions, it does not implement the myriad of computations that are useful for modeling dMRI data. Indeed, there could be many other questions to ask with these data that cannot be addressed with the functionality supported by AFQ-Browser. To facilitate flexible online computation on the data without having to download it to a personal computer, AFQ-Browser closely integrates with Binder, a web-service that allows users to access a cloud-based computational environment that runs a collection of Jupyter notebooks (https://elifesciences.org/labs/8653a61d/introducing-binder-2-0-share-your-interactive-research-environment). Jupyter Notebooks are web applications that store code, text, and figures side by side[67] and run Python (and other language) code through a web-browser user interface. Binder allows users to access collections of notebooks that are stored in Github, and to execute the code in these notebooks through their browser, without having to download any software. To integrate between AFQ-Browser and Binder, the software automatically generates a button in the AFQ-Browser website (Fig. 6, upper right corner) that, upon clicking, directs the visitor's web-browser to a Binder website that contains the data from this AFQ-Browser instance, and an example notebook that reads the data from this website. The example notebook we include performs some data visualization and simple unsupervised learning. Visitors can then extend the code from this example to capitalize on the wealth of statistics, machine learning, and data visualization libraries in Python. Thus, even without downloading the data, AFQ-Browser enables any computation that a visitor can imagine on the shared data, and computations that annotate the data (e.g., adding new labels to the metadata table) can be updated in AFQ-Browser.

**Installation of AFQ-Browser**. AFQ-Browser is open source and distributed under the permissive BSD 3-Clause License, which allows the software to be freely used and redistributed. It is not encumbered by any pending or obtained patents that would limit its use or redistribution. The current version of AFQ-Browser can be cloned from the GitHub repository: https://github.com/YeatmanLab/AFQ-Browser. The current stable release (v0.2) can be found on the Python Package Index (https://pypi.python.org/pypi/AFQ-Browser) and it can be installed, together with all of its dependencies, on any machine with Python and the pip package manager simply by calling: pip install AFQ-Browser. The software includes a test suite implemented using pytest (https://docs.pytest.org/en/latest/), which contains unit testing of portions of the Python code. These tests are also automatically run through a continuous integration system (https://travis-ci.org) for every change introduced to the code.

**Data availability**. The main software described in this study is available through Github at: https://github.com/YeatmanLab/AFQ-Browser/. The version of the software described in this study is also available at the following https://doi.org/10.5281/zenodo.1134626. Code used to generate the figures in this study is available at: https://github.com/yeatmanlab/AFQ-Browser_data (and deposited at the following https://doi.org/10.5281/zenodo.1161846). In addition, we refer to three websites/datasets generated with the software, the source code and data for these websites is available at: https://github.com/yeatmanlab/AFQBrowser-demo/ (https://doi.org/10.5281/zenodo.1161862), https://github.com/yeatmanlab/AFQ-Browser-MSexample/ (https://doi.org/10.5281/zenodo.1161844), and https://github.com/yeatmanlab/Sarica_2017 (https://doi.org/10.5281/zenodo.1161864).

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

## Acknowledgements

The work was funded through a grant by the Gordon & Betty Moore Foundation and the Alfred P. Sloan Foundation to the University of Washington eScience Institute. The work of A.R.-H. was supported by the Department of Energy Computational Science Graduate Fellowship Program of the Office of Science and National Nuclear Security Administration in the Department of Energy under contract DE-FG02-97ER25308. We would like to thank Jeff Heer for providing the original impetus for this work as an assignment in his class on data visualization. We thank Parmita Mehta and Zac Lin for their work on the prototype of AFQ-Browser, Anastasia Yendiki for assistance with TRACULA data formatting, and Ben Cipollini and Roey Schurr for comments on the manuscript. Finally, we would like to thank the authors who contributed the public datasets discussed in this study: Sarica A., Cerasa A., Valentino P., Trotta M., Barone S., Granata A., Nisticò R., Perrotta P., Pucci F., Quattrone A., Mezer A., Wandell B. A.

## Author contributions

J.D.Y., A.R.-H., J.K.S. and A.R. conceived the idea and designed the tool; A.R.-H., J.K.S., A.K. and A.R. wrote the code; J.D.Y., A.R.-H., J.K.S., A.K and A.R. wrote the manuscript.

## Additional information

**Competing interests:** The authors declare no competing interests.

