## [Peer Review File · Nature Communications]

Reviewers' comments:

Reviewer #1 (Remarks to the Author):

In "AFQ-Browser: Supporting reproducible human neuroscience research through browser-based visualization tools," the authors describe their new add on to the AFQ Matlab program, the "AFQ-Browser." This tool packages diffusion tensor imaging data in a tidy format (simple csv tables) and provides a web browser interface to the data to allow secondary analysis. The data and website are automatically deployed to github so that standalone links can be disseminated along with publications or separately. The bulk of the manuscript describes use cases in which secondary analyses of data can reveal hypothesis generation opportunities that would not be pragmatic to include in standard publications. The stated criteria for publication in Nature Communications are:

Criteria: The data is technically sound

Evaluation: The paper and supplementary material appear sound, but I do not have the experience with AFQ to actually test the code. No specific methods are details and no specific results are claimed with regards to the technical superiority of the method relative to another.

- The paper provides strong evidence for its conclusions

The conclusions from the paper are relatively general / non-specific. Hence, they are supported. This paper presents a description of a quality software implementation. It does not present a comparative analysis or technical innovations that support the creative processes and/or specific innovations beyond the union of multiple existing concepts into a well designed package.

- The results are novel (we do not consider abstracts and internet preprints to compromise novelty)

The existence of the AFQ browser package is novel. However, the novelty of the process that led to its creation is not clear. Aside from talent and motivation (both of which the authors clearly have), it is not clear if this manuscript contributes to the body of knowledge that enables the types of programs described.

- The manuscript is important to scientists in the specific field

A major factor limited excitement of the community regarding this package is the relatively low use of AFQ. Based on google scholar, the AFQ paper has been cited 149 times. For comparison, DTIStudio has been cited 833 times while FSL's DTI package has been cited at least 1936 times.

From the text of the manuscript, it does not appear that AFQ Browser would be of use to the non-AFQ users in the DTI community.

Reviewer #2 (Remarks to the Author):

This is a signed review by Chris Gorgolewski

The paper entitled "AFQ--Browser: Supporting reproducible human neuroscience research

through browser--based visualization tools" is a beautifully written description of a software tool that takes outputs a specific of a specific diffusion MRI analysis method (AFQ) and creates interactive visualizations that make data exploration easy. The tool implements some truly innovative ideas such as piggy backing on GitHub as a service for hosting data and visualizations and representation of data in a form that is appealing to data scientists with no prior MR experience. I hope that other tools will emulate those features. The manuscript also includes thoughtful discussion of exploratory vs hypothesis driven methods.

Specific comments:

- The abstract gives the reader the wrong impression that the AFQ-Browser tool is more generic than it really is. It should be clarified that the tool only allows users to visualize and share outputs of AFQ analyses.
- When describing BrainBrowser and its involvement in MACACC dataset surely you meant "visualization" not "analysis".
- It might be worth to introduce the publication feature earlier in the paper. I was quite confused when reading about reproducibility and data sharing without knowing that AFQ-Browser is not just a visualization tool.
- Please mention in the paper the license under which the tool is distributed and any pending or obtained patents that would limit its use or redistribution.
- If all AFQ users start uploading their results to GitHub using AFQ-Browser it might be hard to find or aggregate those results. It might be worth considering (and discussing) a centralized index (also hosted on GitHub) of all publicly available AFQ-Browser generated bundles. This index can be automatically updated during the "publish" procedure.
- GitHub is a great resource, but have few guarantees in terms of long term storage. A solution to this would be depositing the bundles into Zenodo which could be done directly from GitHub. Would be worth implementing and/or discussing this in the manuscript.
- It's a technical detail, but it took me a little time to figure out why the tool requires user to spin up a local server (presumably to be able to access CSV and JSON files). Might be worth elaborating.
- Saving the visualization "view" (or "browser state") seems cumbersome when done via a file. Could the view be encoded in the URL (via GET parameters)? Sharing of such views would be much easier and natural.
- Some example analyses include information about group membership or demographic information such as age. How is such information stored and conveyed to AFQ-Browser? Does it also come as output of AFQ?
- In the manuscript you mention that AFQ-Browser allows users to compare their results with normative distributions. Where are they coming from a central repository (please describe how it is populated) or do users need to provide such distributions themselves?
- It might be worth considering a crowdsourcing scheme such as the one employed in MRIQC Web API (<https://mriqc.nimh.nih.gov/>) to generate normative distributions of AFQ outputs.
- Is the way you store data in CSV files and their relation to the JSON files (beyond the "tidy" convention) described somewhere in detail? It would be useful for users.
- Please describe the software testing approach you employed in this project.

Reviewer #3 (Remarks to the Author):

- Major claims

This paper presents AFQ browser, which allows the user to visualize tract-level statistics (FA, MD etc) for each of the 20 major tracts analyzed/produced by AFQ. The visualization is done both at group level (mean, sd/se) and at the individual subject level. Tools such as brushable tracts allow for more precise examination of subregions of tracts. Linked visualization will simultaneously display segments of the tracts for stats and anatomy. Linked visualization can also link each curved under examination and specific subject.

The authors claim that this level of detail and cross-link can aid reproducible science. Several examples are given: data published in previous papers can be used in AFQ browser to identify individual subjects; specific segments of specific tracts can be identified to be particularly vulnerable to particular diseases (e.g., CST for ALS).

Another major claim is that any lab can publish the interactive AFQ browser website for their own data by a simple `afqbrowser-publish` command and use of GitHub, and that this will greatly facilitate sharing of visualizable data and allow for more analyses based on the data.

AFQ browser software is therefore browser-based visualization and analysis of white matter tracts that is "one-click" publishing results to a free GitHub pages website.

- Novelty

The authors themselves list a number of similar browser-based visualization tools, but they argue compellingly that their method offers a distinct advantage. Being able to visualize and manipulate individual-based data from AFQ outputs is novel. AFQ browser also fills an important gap where individual researchers without access to programming expertise or manpower can use the website created by `afqbrowser-publish` to visually interact with published data directly.

- Interest to the field

The paper will be of interest to others in the field. For example, researchers who generate and who use large amounts of AFQ-like white matter properties, researchers who are interested in individual-level questions, will be interested in reusing the long-tail of unpublished data made available this way.

- Moving the field

This is one of the many tools in the field that reflect the current trend of big-data science. Publishing data accompanying publications has received a lot of enthusiasm from the field.

However, mechanisms for this have not been ideal. The authors present an intermediate means that is to publish derived, intermediate data, using a free website visualization approach. This makes publication of data more feasible and more reusable. In this way, the paper can move the field with respect to publishing useful intermediate data.

Authors claim that AFQ browser will facilitate deep-data science. While data can be downloaded and analyzed offline (but then one does not need AFQ browser for it), the amount of data that can be linked to individual tracts are small and not manipulable within the browser. Authors present a simple (albeit elegant) interface for interacting with data. More complex ways of interaction with the data would improve the potential to influence the thinking in the field.

- Improvements

AFQ browser is a great interactive visualization tool. Authors claim that re-using intermediate data such as what AFQ browser uses from AFQ can lead to more reproducible research, vs. using raw data. The view point is debatable. The AFQ outputs – diffusion property measures already computed along each of the 20 major tracts – can be submitted to traditional statistical/machine learning analyses as authors suggest. Authors have provided several examples where analyses were done to extend on published studies. However, those examples are about doing additional analysis, which is not exactly reproducing previously reported findings. It is difficult to imagine a scenario where these input data would not reproduce previously published results. Authors should provide examples of reproducibility, and why AFQ browser uniquely allows for that (i.e., without AFQ browser, reproducibility would be low).

Such visualization would be more appealing if it can interact with other popular DTI software packages (FSL, DTI studio etc), for example by making the platform open source.

Authors do not mention whether the segments visualized by “brushable tracts” can be indicated – i.e., which sets of nodes correspond to the shaded brush so that that part of the data can be separately analyzed.

Adding new subjects whose data are collected and analyzed in a different manner should be done with caution. Authors should devise a way to deal with this.

Authors emphasized that one advantage of AFQ browser is that it allows individual subject data to be visualized, and that this may be particularly important for surgery, example. While it is true that individual subject tract and metadata can be selected and examined, in order to provide validity that an individual subject is different from group mean, some statistics such as tract, brushing or node-level z-score should accompany each selected subject. It is not clear how the z-scores that authors quoted (e.g., text for Figure 4) were obtained or ascertained on AFQ browser.

AFQ-Browser Revision: Point-By-Point Response to Reviews

We would like to thank the reviewers for the insightful comments on this work. Based on themes that were consistent in the three reviews, we have completed a substantial revision of the AFQ-Browser code base, and released a new version of AFQ-Browser (v0.2). We begin by describing three major changes that are relevant to all the reviewers. In addition to these major revisions, we have also made a series of improvements to the code and manuscript to address each of the reviewers' individual comments. The **reviewers' comments are in red**, and our responses and specific actions are detailed in line **in black**. We have tried to document these changes as pull requests and issues in the AFQ-Browser github repository, and we provide links where appropriate for technical details on the changes. We agree with the reviewers' assessment that these changes will substantially increase the impact of this work.

Major revisions to the software and manuscript

1. **Compatibility with a broad range of diffusion MRI analysis packages.** All three reviewers agreed that this work would have greater impact if AFQ-Browser was broadly applicable to diffusion MRI researchers who might not use AFQ. We have undertaken a major revision of the AFQ-Browser code base to make this possible and provide two solutions. **First**, we have added support for FSL users by building an automated pipeline to spin up an instance of AFQ-Browser for data processed with FSL and TRACULA (See issue #163, pull requests #181, #200, #209). FSL is the most widely used toolbox for processing diffusion MRI data (based on citations), and TRACULA is an automated pipeline for conducting a tract-based analysis of diffusion data processed with FSL. In short, we have modified the `afqbrowser-assemble` function to be able to create an AFQ-Browser website based on the data output from this alternative pipeline. We have also added new rendering capabilities to the 3D ANATOMY window to support their tractography format. Thus, AFQ-Browser is now fully automated for FSL users who wish to analyze and publish data using AFQ-Browser. **Second**, we have added extensive documentation and data specifications to AFQ-Browser, and provided a simple API for other developers to leverage this tool. Now, data from any software package that implements a tract-based analysis could be visualized and published with AFQ-Browser, so long as the data is properly organized. This documentation is now automatically generated from the source code and can be found at: <https://yeatmanlab.github.io/AFQ-Browser/> (See Issue #164 and pull requests #172, #220, #221)
2. **AFQvault - A centralized database for AFQ-Browser.** All three reviewers agreed that the long-term impact of AFQ-Browser would be much greater if there was a centralized repository that stored data from every published AFQ-Browser instance. We fully agree with this point - as more researchers publish data through AFQ-Browser, aggregating data in a centralized database will lead to powerful new discoveries that leverage this public resource. To accomplish this we have designed a custom, queryable database system for AFQ-Browser that is hosted at <http://afqvault.org> (See Issue #166). The `afqbrowser-publish` function now automatically submits data to this centralized database after building the AFQ-Browser website on the user's github page (See pull request #208). Homogenizing data acquired on different scanners, and with different forms of

pre-processing, is an ongoing challenge in diffusion MRI research. To support this goal, the afqvault database stores all the metadata and diffusion MRI acquisition parameters associated with the data (see Issue #170). Thus anyone will be able to search the afqvault database for datasets with specific combinations of parameters or metadata characteristics, combine these data, and then analyze the effects of specific scan sequences or metadata characteristic.

- 3. Binder integration:** Reviewers all expressed an interest to see AFQ-Browser do more in terms of its analytical functionality. While we are reluctant to overload the browser-based visualization with a lot of additional knobs and buttons that would allow more flexibility in terms of the analysis, we appreciate that users of the software might want to immediately explore the data-sets further (e.g., with machine learning), without the need to download the data to their own computers. In the revised version of the software and manuscript, we were able to take advantage of an innovative new cloud-based computing system called Binder, that allows researchers to package data and analysis for others to not only explore, but expand with their own ideas. (see: <https://elifesciences.org/labs/8653a61d/introducing-binder-2-0-share-your-interactive-research-environment>). The new versions of AFQ-Browser websites now include a button that will take visitors to a Binder website where the data is already available and can be explored using Python code executed in a Jupyter notebook. Importantly, because it runs in the cloud, this system requires no additional software installation or any data download, and at the same time allows visitors to the site to explore the data with whatever analysis they can imagine. Standard data science software tools, such as Pandas and scikit-learn are already installed into the Binder site for visitors to use. We provide an example notebook that implements some simple unsupervised learning algorithms on these data. See pull request #213 and #223.

Reviewer #1 (Remarks to the Author):

A major factor limited excitement of the community regarding this package is the relatively low use of AFQ. Based on google scholar, the AFQ paper has been cited 149 times. For comparison, DTIStudio has been cited 833 times while FSL's DTI package has been cited at least 1936 times.

From the text of the manuscript, it does not appear that AFQ Browser would be of use to the non-AFQ users in the DTI community.

We appreciate this feedback and have completed a substantial revision to the code base and documentation to make sure that AFQ-Browser can be integrated with any diffusion MRI package that implements a tract-based analysis. Specifically we have built an automated interface between FSL and TRACULA and AFQ-Browser. Since DTIStudio does not have an automated analysis pipeline, it is not feasible for us to automate the process of ingesting DTIStudio data: Every user might organize their data in a different way and manually define a different collection of tracts. However, by simplifying the data specifications, and writing extensive documentation <https://yeatmanlab.github.io/AFQ-Browser/dataformat.html> , we have ensured that users of DTIStudio (and any other package) can take advantage of AFQ-Browser

with minimal additional coding. The *afqbrowser-assemble* and *afqbrowser-publish* functions are now flexible enough to handle data from a broad array of software packages as long as users/developers adhere to the defined data specifications.

Reviewer #2 (Remarks to the Author):

This is a signed review by Chris Gorgolewski

The paper entitled “AFQ-Browser: Supporting reproducible human neuroscience research through browser-based visualization tools” is a beautifully written description of a software tool that takes outputs of a specific diffusion MRI analysis method (AFQ) and creates interactive visualizations that make data exploration easy. The tool implements some truly innovative ideas such as piggy backing on GitHub as a service for hosting data and visualizations and representation of data in a form that is appealing to data scientists with no prior MR experience. I hope that other tools will emulate those features. The manuscript also includes thoughtful discussion of exploratory vs hypothesis driven methods.

The abstract gives the reader the wrong impression that the AFQ-Browser tool is more generic than it really is. It should be clarified that the tool only allows users to visualize and share outputs of AFQ analyses.

We have taken this point seriously and completed a substantial revision of the code that, first, makes AFQ-Browser compatible with other widely used diffusion MRI analysis tools and, second, documents the data specifications such that other developers could easily leverage AFQ-Browser for visualizing and sharing the analysis results from other diffusion MRI pipelines that we have not considered. We describe these revisions above and provide links to the relevant github pull requests. Additionally, we have revised the manuscript to note how AFQ-Browser interacts with other analysis tools.

When describing BrainBrowser and its involvement in MACACC dataset surely you meant “visualization” not “analysis”.

Thank you for pointing this out. We have revised the text on page 2 of the Introduction to more accurately describe BrainBrowser.

It might be worth to introduce the publication feature earlier in the paper. I was quite confused when reading about reproducibility and data sharing without knowing that AFQ-Browser is not just a visualization tool.

Thank you for the suggestion. To make this point more apparent, we now introduce the idea of publishing visualizations and underlying data to the web in the first paragraph. Moreover, we have added a sentence to the **Scientific reproducibility** section of the Introduction which introduces the publication feature.

Please mention in the paper the license under which the tool is distributed and any pending or obtained patents that would limit its use or redistribution.

In the Methods section “Installation of AFQ-Browser”, we now note the license and AFQ-Browser does not have any patents that would limit distribution:

AFQ-Browser is open source and distributed under the permissive BSD 3-Clause License, which allows the software to be freely used and redistributed. It is not encumbered by any pending or obtained patents that would limit its use or redistribution

To clarify that the tool is available for anyone to use, we have also added the words “open source” to the abstract.

If all AFQ users start uploading their results to GitHub using AFQ-Browser it might be hard to find or aggregate those results. It might be worth considering (and discussing) a centralized index (also hosted on GitHub) of all publicly available AFQ-Browser generated bundles. This index can be automatically updated during the “publish” procedure.

We would like to thank the reviewer for making this suggestion. In addition to providing a simple means for scientists to publish their data, it is also important to begin aggregating these data in a centralized location. We have made three major improvements to the AFQ-Browser package in response to the reviewer’s suggestion (described on page 1 of this response and reviewed here).

1. In addition to publishing the data to a github repo owned by the user, afqbrowser-publish now also commits a link to the data in a centralized repository: <https://github.com/afqvault/afqvault>. Thus, there is a record of every published AFQ-Browser site (see issue #166 and pull request #208)
2. We have implemented a centralized database at <http://afqvault.org>. Using a continuous integration system (Travis CI), data from each published AFQ-Browser site is automatically committed into this centralized database once we accept the pull request against the afqvault repo. This central database stores the raw data associated with each AFQ-Browser instance..
3. The centralized afqvault database also stores scan parameters and other metadata characteristics associated with the metadata. This will make it possible to properly curate the data in the long run, though curation and meta-analysis is well beyond the scope of the present manuscript. See issues #170 and pull request #225.

GitHub is a great resource, but have few guarantees in terms of long term storage. A solution to this would be depositing the bundles into Zenodo which could be done directly from GitHub. Would be worth implementing and/or discussing this in the manuscript.

We agree with the reviewer that ensuring long-term storage of the data is important. We didn't want to force every user to get an account with Zenodo, so the solution we implement capitalizes on the new database we have built (afqvault). When a user publishes their data to github with afqbrowser-publish, it also forks the main afqvault repository, commits a link to the new github page, and submits a pull request. When the pull request is merged, the data is entered into the afqvault database. We would certainly like for users to create DOI pointing to their data, but we would like users to have control over this process. To facilitate this, we have created a documentation web page that explains how users might do this (see pull request #252 and). Additionally, we have added a discussion of this issue in the manuscript:

Long-term preservation of the data is important and, because GitHub does not guarantee long-term storage, we suggest using another service to ensure that the data is accessible in perpetuity. A one-click solution is provided through Zenodo (<http://zenodo.org/>)³⁴, a website developed by CERN specifically to support long-term preservation of data and other research products. Zenodo can be used to mint a persistent digital object identifier (DOI) for GitHub repositories. Other solutions include institutional repositories, to which users of AFQ-Browser can upload their data. We do not intend to enforce one solution or another, and provide users with maximal control over this process. However, to provide users with information about the reasons to pursue long-term preservation, and about one way they could approach this, we provide documentation for issuing a DOI through Zenodo (https://yeatmanlab.github.io/AFQ-Browser/long_term_preservation.html).

It's a technical detail, but it took me a little time to figure out why the tool requires user to spin up a local server (presumably to be able to access CSV and JSON files). Might be worth elaborating.

Yes, this is correct. We have added a sentence to the Methods section of the paper to explain this:

A second command line function, afqbrowser-run, launches a static web-server on the user's computer with AFQ-Browser running for this dataset. Running a static web-server is required to access locally stored data files.

Saving the visualization “view” (or “browser state”) seems cumbersome when done via a file. Could the view be encoded in the URL (via GET parameters)? Sharing of such views would be much easier and natural.

This is a a great suggestion and solves many problems with the settings file. We now save the browser state in a query string. This means that only one AFQ-Browser website needs to be created for a dataset. Multiple browser states can then be saved and cited based on copying or

bookmarking the URL once the desired set of parameters has been determined. The technical details can be seen in pull request #185. We describe this new feature in the Methods:

Reproducing results that are generated by a graphical user interface (GUI) can be problematic since figures are generated based on a series of user inputs (i.e., mouse clicks and key presses). To solve this problem, we record the series of interactions as a query string, and append the URL with each user input to AFQ-Browser. By copying or bookmarking the URL, a user can save and re-open a specific state of AFQ-Browser. Hence, a discovery made through a series of operations in the GUI can be communicated (and reproduced) without a lengthy description of the series of user inputs.

Some example analyses include information about group membership or demographic information such as age. How is such information stored and conveyed to AFQ-Browser? Does it also come as output of AFQ?

The AFQ software already includes a convention for storing metadata (See AFQ wiki <https://github.com/yeatmanlab/AFQ/wiki#including-subject-metadata-in-the-afq-structure>). But metadata can now also be added to AFQ-Browser with a .csv file (see pull request #181). Any column in the AFQ-browser/client/data/subjects.csv file is added to the metadata table in AFQ-Browser. So it is easy to add new metadata from a spreadsheet to AFQ-Browser if it was not included in the original AFQ analysis. This also makes it simple for users of other DTI software packages (e.g., FSL) to add metadata to AFQ-Browser. This data format is also documented in <https://yeatmanlab.github.io/AFQ-Browser/dataformat.html>

In the manuscript you mention that AFQ-Browser allows users to compare their results with normative distributions. Where are they coming from a central repository (please describe how it is populated) or do users need to provide such distributions themselves?

The comparison between patients and a normative distribution requires the user to supply control subjects. For example in Figure 4 and Figure 5, the dataset included a patient and a control group. By selecting the column of the metadata table indicating disease state, and then selecting standard deviation from the Tract Details control panel, each patient is plotted against the normative distribution. As suggested by the reviewer, we have implemented a centralized repository with a queryable database such that (in the future) we can supply normative distributions that are matched to any specific sample. However, this goal depends on developing methods to homogenize data collected at different sites, and with different scanners. Combining diffusion data for multi-site studies is an important and active area of research that we hope to contribute to by curating and sharing this centralized afqvault database. But for the time being, users must collect their own control data.

It might be worth considering a crowdsourcing scheme such as the one employed in MRIQC Web API (<https://mriqc.nimh.nih.gov/>) to generate normative distributions of AFQ outputs.

Thank you for this suggestion. We have capitalized on MRIQC to develop the centralized afqvalut database so that this will be possible in the future. As noted by Reviewer 3, we will have to be cautious about combining data across sites, but we believe that this is a surmountable challenge once afqvault has aggregated a sufficient amount of data.

Is the way you store data in CSV files and their relation to the JSON files (beyond the “tidy” convention) described somewhere in detail? It would be useful for users.

In the revision of the code base, we have included detailed documentation of the data-format (<https://yeatmanlab.github.io/AFQ-Browser/dataformat.html>).

Please describe the software testing approach you employed in this project.

Python components of the software are unit tested with the pytest framework and continuous integration is implemented using Travis. This means that Python components are exercised with a rudimentary set of tests, that are automatically executed for every proposed change in the code.

Reviewer #3 (Remarks to the Author):

This is one of the many tools in the field that reflect the current trend of big-data science. Publishing data accompanying publications has received a lot of enthusiasm from the field. However, mechanisms for this have not been ideal. The authors present an intermediate means that is to publish derived, intermediate data, using a free website visualization approach. This makes publication of data more feasible and more reusable. In this way, the paper can move the field with respect to publishing useful intermediate data.

Authors claim that AFQ browser will facilitate deep-data science. While data can be downloaded and analyzed offline (but then one does not need AFQ browser for it), the amount of data that can be linked to individual tracts are small and not manipulable within the browser. Authors present a simple (albeit elegant) interface for interacting with data. More complex ways of interaction with the data would improve the potential to influence the thinking in the field.

The reviewer brings up an important point: Ultimately, the potential for Big Data to have a transformative impact on the field is not in conducting group comparisons with better statistical power, but in testing, applying and validating more complex models that would not be possible on a small dataset. However, interactive data visualization, with links between different views of the data, is an efficient way to explore and understand relationships that are present in a high-dimensional dataset before developing a formal model. The goal of AFQ-Browser is to facilitate this exploratory phase of data analysis, and we have written the software to run efficiently in a web-browser **without any server side computations**. The decision to have all the analysis

done client-side means that a user can rapidly explore a dataset without needing to pay fees for cloud-compute services (e.g., Amazon Web Services), and removes barriers that these pay-per-use services impose (e.g., making an account on AWS, providing billing information, keeping track of monthly fees). We feel that the ease that comes with a fully client-side application is one of the major benefits of AFQ-Browser, and was carefully considered as we designed this tool. However, this decision also precludes implementing more complex analyses directly within AFQ-Browser.

But we do also agree with the reviewer on the importance of helping users segue from exploratory data analysis, to formal statistics and modeling of the data. We have implemented a solution that capitalizes on a new service, Binder. This service allows users to compile a website that contains Jupyter notebooks that can be executed in the browser, from a GitHub repository. We have connected AFQ-Browser with Binder, such that a Binder instance becomes automatically available, when a user of AFQ-Browser publishes their website on GitHub. The revised version of the published AFQ-Browser instances now contain a button to launch Binder directly from AFQ-Browser, allowing a visitor to the AFQ-Browser instance to implement any type of computation (e.g., machine learning or unsupervised learning) directly on the data in AFQ-Browser in the web browser. When a user clicks this button, it will open a Jupyter notebook that already includes examples of a series of more complex computations and visualizations of the data. The user can easily edit this notebook to meet their specific needs and update the visualization in AFQ-Browser. We describe this new feature in the Methods section:

Integration of AFQ-Browser and Binder to support extensible computations

While AFQ-Browser supports flexible exploration of the data over many dimensions, it does not implement the myriad of computations that are useful for modeling dMRI data. Indeed, there could be many other questions to ask with these data that cannot be addressed with the functionality supported by AFQ-Browser. To facilitate flexible online computation on the data without having to download it to a personal computer, AFQ-Browser closely integrates with Binder, a web-service that allows users to access a cloud-based computational environment that runs a collection of Jupyter notebooks (<https://elifesciences.org/labs/8653a61d/introducing-binder-2-0-share-your-interactive-research-environment>). Jupyter Notebooks are web applications that store code, text and figures side-by-side⁴³ and run Python (and other language) code through a web-browser user interface. Binder allows users to access collections of notebooks that are stored in Github, and to execute the code in these notebooks through their browser, without having to download any software. To integrate between AFQ-Browser and Binder, the software automatically generates a button in the AFQ-Browser website that, upon clicking, directs the visitor's web-browser to a Binder website that contains the data from this AFQ-Browser instance, and an example notebook that reads the data from this website. The example notebook we include performs some data visualization and simple

unsupervised learning. Visitors can then extend the code from this example to capitalize on the wealth of statistics, machine learning and data visualization libraries in Python. Thus, even without downloading the data, AFQ-Browser enables any computation that a visitor can imagine on the shared data, and computations that annotate the data (e.g., adding new labels to the metadata table) can be updated in AFQ-Browser.

AFQ browser is a great interactive visualization tool. Authors claim that re-using intermediate data such as what AFQ browser uses from AFQ can lead to more reproducible research, vs. using raw data. The view point is debatable. The AFQ outputs – diffusion property measures already computed along each of the 20 major tracts – can be submitted to traditional statistical/machine learning analyses as authors suggest. Authors have provided several examples where analyses were done to extend on published studies. However, those examples are about doing additional analysis, which is not exactly reproducing previously reported findings. It is difficult to imagine a scenario where these input data would not reproduce previously published results. Authors should provide examples of reproducibility, and why AFQ browser uniquely allows for that (i.e., without AFQ browser, reproducibility would be low).

In the original manuscript we argue that AFQ-Browser will lead to more reproducible research by removing barriers to sharing data alongside a publication. It is important to note that data sharing is a critical component of reproducible research but has not yet become common practice. Without tools like AFQ-Browser it can be difficult for some scientists to figure out easy ways to share data. Meanwhile, providing a carrot in the form of an elegant data visualization also provides the much-needed additional incentive that prevents researchers from sharing data in many cases.

We acknowledge that the previous version of the manuscript did not sufficiently delineate what exactly we mean by “reproducibility”, and this needs to be clarified, as there is some confusion among scientists about the meaning of reproducibility (Baker, 2016. 1,500 scientists lift the lid on reproducibility. *Nature*, 533(7604), 452–454; Baker, 2016. Muddled meanings hamper efforts to fix reproducibility crisis. *Nature News*.). We emphasize that we follow here the definition reproducibility in the sense discussed in by Goodman, Fanelli, & Ioannidis (2016):

“the ability of a researcher to duplicate the results of a prior study using the same materials as were used by the original investigator. That is, a second researcher might use the same raw data to build the same analysis files and implement the same statistical analysis in an attempt to yield the same results.”

We have revised the manuscript text to refer readers to recent papers that are intended to provide a definitive definition of reproducibility for the field. In the Introduction we now write:

The field of human neuroscience faces several specific challenges with regards to reproducibility (see ^{25,26} for a definition of reproducible research).

The distinction between reproducibility and replicability is the most widely used in the biological sciences and appears in the wikipedia page on reproducibility:

“Reproducibility is the ability to get the same research results using the raw data and computer programs provided by the researchers. A related concept is replicability, meaning the ability to independently achieve similar conclusions when differences in sampling, research procedures and data analysis methods may exist. Reproducibility and replicability together are among the main principles of the scientific method (<https://en.wikipedia.org/wiki/Reproducibility>)”

We re-iterate this distinction here and in the manuscript because the reviewer states that “It is difficult to imagine a scenario where these input data would not reproduce previously published results”, but our point is that **without access to the previously published data it is impossible to reproduce previously published results**. AFQ-Browser is designed to facilitate data access and thus facilitate reproducibility. Of course reproducibility does not ensure that a study will be *replicated*. But if published findings can't be reproduced, then it is much more difficult to trust the conclusions or conduct a replication.

Additionally, to clarify this point, we have amended the analysis of data reported in Sarica et al. 2017 to specifically address the reviewer's point. Sarica and colleagues made this data publicly available by using the `afqbrowser-publish` function, and the Jupyter notebook now cited in the Results section of the manuscript reproduces exactly a figure from their manuscript based on the data in AFQ-Browser. This would not be possible without data sharing and would be much more difficult if working from the raw data.

Such visualization would be more appealing if it can interact with other popular DTI software packages (FSL, DTI studio etc), for example by making the platform open source.

The AFQ-Browser software is open source and available in an open github repository (<https://github.com/yeatmanlab/AFQ-Browser>). We apologize if this was not clear in the original manuscript. We have added the words “open source” to the abstract, to make that clear. In order to facilitate broader adoption of this platform, we have: (1) Added support for researchers using other DTI software packages including FSL/TRACULA (See “Major revisions to the software and manuscript” and issue #163 and pull request #181 in the github repository); (2) Written extensive documentation for the software to make it easier for other developers to incorporate the open-source code into their projects and develop APIs for other software packages to directly interact with AFQ-Browser (see pull request #220). This new documentation is built as a website that lives within the AFQ-Browser repository: <https://yeatmanlab.github.io/AFQ-Browser/>; see issue #164). Thus, AFQ-Browser can now automatically be used to visualize, analyze and share results of diffusion MRI analyses conducted in multiple software packages and it can be easily extended (with just a few lines of code) to interface with any other package that adheres to a few standard conventions. Without

any coding, users of any software package can now employ AFQ-Browser to visualize and share data as long as they follow our documented specifications for organizing data.

Authors do not mention whether the segments visualized by “brushable tracts” can be indicated – i.e., which sets of nodes correspond to the shaded brush so that that part of the data can be separately analyzed.

Thank you for the suggestion. We have added this new feature to the revised code base. “Brushable tracts” now displays the node numbers so that a user can conduct statistical analyses of this region of the tract outside of AFQ-Browser. For example, in the below figure, it indicates that the selection goes from node 12-51. See issue #165 and pull request #175.

Adding new subjects whose data are collected and analyzed in a different manner should be done with caution. Authors should devise a way to deal with this.

We would like to thank the reviewer for raising this point. While there has been substantial progress in developing algorithms to homogenize diffusion MRI data collected on different scanners or with different pulse sequences, naively assuming that data can be combined across studies can lead to many problems. However, the development of effective methods to combine and compare data collected at different sites will be catalyzed by the increasing number of publicly available datasets supported by AFQ-Browser. In response to Reviewer #2 we have built a centralized database to aggregate and store data that is published by AFQ-Browser (<http://afqvault.org>). Data is now automatically committed to this centralized database when a user runs the *afqbrowser-publish* function. As part of this centralized database, we also store parameters associated with the data acquisition (see issue #166 and issue #122). As noted by the reviewer, this additional information will be critical for researchers who wish to combine data across studies.

Authors emphasized that one advantage of AFQ browser is that it allows individual subject data to be visualized, and that this may be particularly important for surgery, example. While it is true that individual subject tract and metadata can be selected and examined, in order to provide validity that an individual subject is different from group mean, some statistics such as tract, brushing or node-level z-score should accompany

each selected subject. It is not clear how the z-scores that authors quoted (e.g., text for Figure 4) were obtained or ascertained on AFQ browser.

We would like to thank the reviewer for this suggestion. We have added this new feature. Now, hovering over a selected subject's data displays the z-score at the selected node relative to the norms for each group. Thus, a patient can be compared against a control group, or relative to other patients in their group. We also display the groupings in case that the user has selected multiple groups. In this case we show z-scores for the individual against each group. See pull request #216. The below figure shows the mouse hover-over for a single subject when 2 groups are selected. This subject has a z-score of 1.15 relative to their own group and 1.49 relative to the other group at the node that the mouse is hovering over.

REVIEWERS' COMMENTS:

Reviewer #1 (Remarks to the Author):

My concerns have been addressed.

Reviewer #2 (Remarks to the Author):

Excellent work.

Reviewer #3 (Remarks to the Author):

Authors have addressed all of the concerns raised by this reviewer.

--Lei Wang